# FastTrack: An open-source software for tracking varying numbers of deformable objects

**Benjamin Gallois, Raphaël Candelier**⊙*

Sorbonne Université, CNRS, Institut de Biologie Paris-Seine (IBPS), Laboratoire Jean Perrin (LJP), Paris, France

* raphael.candelier@sorbonne-universite.fr

## Abstract

Analyzing the dynamical properties of mobile objects requires to extract trajectories from recordings, which is often done by tracking movies. We compiled a database of two-dimensional movies for very different biological and physical systems spanning a wide range of length scales and developed a general-purpose, optimized, open-source, cross-platform, easy to install and use, self-updating software called FastTrack. It can handle a changing number of deformable objects in a region of interest, and is particularly suitable for animal and cell tracking in two-dimensions. Furthermore, we introduce the probability of incursions as a new measure of a movie's trackability that doesn't require the knowledge of ground truth trajectories, since it is resilient to small amounts of errors and can be computed on the basis of an *ad hoc* tracking. We also leveraged the versatility and speed of FastTrack to implement an iterative algorithm determining a set of nearly-optimized tracking parameters —yet further reducing the amount of human intervention—and demonstrate that FastTrack can be used to explore the space of tracking parameters to optimize the number of swaps for a batch of similar movies. A benchmark shows that FastTrack is orders of magnitude faster than state-of-the-art tracking algorithms, with a comparable tracking accuracy. The source code is available under the GNU GPLv3 at https://github.com/FastTrackOrg/FastTrack and pre-compiled binaries for Windows, Mac and Linux are available at http://www.fasttrack.sh.

## Author summary

Many researchers and engineers face the challenge of tracking objects from very different systems across several fields of research. We observed that despite this diversity the core of the tracking task is very general and can be formalized. We thus introduce the notion of *incursions—i.e.* to what extent an object can enter a neighbor's space—which can be defined on a statistical basis and captures the interplay between the acquisition rate, the objects' dynamics and the geometrical characteristics of the scene, including density. To validate this approach, we compiled a dataset from various fields of Physics, Biology and human activities to serve as a benchmark for general-purpose tracking softwares. This

**Data Availability Statement:** All data files are available from the TD2 database (http://data.ljp.upmc.fr/datasets/TD2/).

**Funding:** This work has been financed by an ANR JCJC grant (ANR-16-CE16-0017, https://anr.fr/) received by RC.

**Competing interests:** The authors have declared that no competing interests exist.

dataset is open and accepts new submissions. We also developed a software called *Fast-Track* that is able to track most of the movies in the dataset by proposing standard image processing tools and state-of-the-art implementation of the matching algorithm, which is at the core of the tracking task. Besides, it is open-source, simple to install and use and has an ergonomic interface to obtain fast and reliable results. FastTrack is particularly convenient for small-scale research projects, typically when the development of a dedicated software is overkill.

This is a *PLOS Computational Biology* Software paper.

## Introduction

Tracking objects moving in two dimensions is a demanded ability in computer vision, with applications in various fields of Biology ranging from the monitoring of cellular motion [1, 2] to behavioral assays [3, 4], but also in many other fields of Science—like microfluidics [5], active matter [6], social sciences [7] or robotics [8] to name a few—and in industrial processes [9]. There has been countless libraries and softwares designed for the purpose of tracking specific systems in specific conditions, but to date none has emerged as a general tool that could track virtually any type of object in a large panel of imaging conditions.

This owes for a large part to the many issues arising during the detection phase, *i.e.* the definition of the objects that are present on each frame. In order to lower the error rate of the whole tracking process, many softwares focus on adapting the details of object detection to the system of interest [4]. Yet, in various situations the objects are allowed to partly overlap (quasi two-dimensional systems), making proper detection extremely challenging. In particular, this is very common among biological systems since a strict planar confinement is often difficult to achieve and may bias the object's dynamics [10–12]. A few algorithms have been developed to manage these situations by defining a unique identifier for each object that allows to recombine the trajectory fragments before and after occlusions [13, 14]. This approach is however computationally heavy and limited to a small number of well-defined, non-saturated objects.

Here we take a different approach and provide a software designed to be as general as possible. First, we created an open dataset comprising 41 movies of very different systems in order to test and benchmark general tracking softwares. Then, we created the FastTrack software that implements standard image processing techniques and a performant matching procedure. FastTrack can handle deformable objects as long as they keep a constant area and manages flawlessly a variable number of objects. For the end user, all these features allow to obtain excellent trackings in minutes for a very large panel of systems. To achieve perfect trackings, FastTrack also has a manual post-processing tool that displaces the workload from the fine-tuning of complex detection algorithms to the manual correction of a few remaining errors, if any. This strategy does not require programming skills and with an ergonomic interface it is time-saving for small size datasets or when there is a very low tolerance for errors.

Furthermore, we propose a new quantifier called the probability of incursions, that can be computed based on a statistical analysis of the dynamical and geometric properties of each movie. We show that this probability displays a remarkable scaling with the logarithm of the

sampling timescale, and that one can easily derive a robust and practical *ad hoc* characterization of virtually any movie. Finally, we implemented an algorithm to determine nearly-optimized tracking parameters automatically. FastTrack has already been used in a few publications [15, 16] and is currently used for several research projects in Physics and Biology. The raw data for all plots can be found in S1 Data.

## Dataset

We compiled various video recordings to form a large dataset called the Two-Dimentional Tracking Dataset (TD$^2$). It is open to new contributions and available for download at http://data.ljp.upmc.fr/datasets/TD2/. All videos have been either previously published [13, 15–26] or have been kindly provided by their authors and are licenced under the Non-Commercial, Share-Alike Creative Commons licence (CC BY-NC-SA). Each movie has an unique identifier composed of three letters and three digits (*e.g.* ACT_001), that we use in the sequel anytime we need to refer to the data.

The dataset comprises 41 sequences involving different types of objects at various scales: bacteria, paramecia, cells in a dense tissue, 7 animal species (including some whose behavior is commonly studied: fruit flies, medaka, zebrafish and mice), self-propelled particles, passive hard particles, droplets in microfluidic channels, centimetric robots, macroscopic objects on a conveyor belt, humans playing sports and traffic. A summary of the key features of each movie in the dataset is presented in S1 Table, thumbnails of the dataset are show in S1 Fig and S1 Video is a footage of the movies with the trajectories overlaid.

The number of objects is constant in about half of the movies (22), while in the other half some objects appear or disappear at various locations in the field of view. Independantly, in about half the movies (20) the objects are moving in a strict two-dimensional space, while for the other half the objects evolve in a quasi-2D space and can at least partly overlap on some frames.

## Software design and implementation

FastTrack is written in C++ and respects the object-oriented programming paradigm. We use the OpenCV library for image processing and the Qt framework for the graphical user interface. The software wokflow is depicted in Fig 1, and can be broken down in three main phases: detection, matching and post-process.

As an entry point, the user defines data sources as video files (all formats supported by FFmpeg) or sequences of frames (`*.bmp`, `*.dib`, `*.jpeg`, `*.jpg`, `*.jpe`, `*.jp2`, `*.png`, `*.pbm`, `*.pgm`, `*.ppm`, `*.sr`, `*.ras`, `*.tiff` or `*.tif`) in a folder; in the latter case the images must have a naming convention with left-padded zeros to ensure a correct ordering of the frames (*e.g.* `frame_000001.pgm`). The whole process described below can be applied on a movie-per-movie basis or for a batch of movies. In the latter case, the user can define different sets of parameters and background images, either manually or *via* a configuration file, and select subsets of movies for which the software will use those sets.

### Detection

This phase aims at extracting a collection of kinematic parameters (*e.g.* positions, direction, area) for each object in a frame. FastTrack includes a collection of standard image processing techniques to let the user adjust and perform object detection within the graphical interface for simple movies, without the need of an external pre-processing.

**Image registration.** It is common to have translational and rotational drifts in movies, and several registration options are available to compensate for it [27]. FastTrack implements

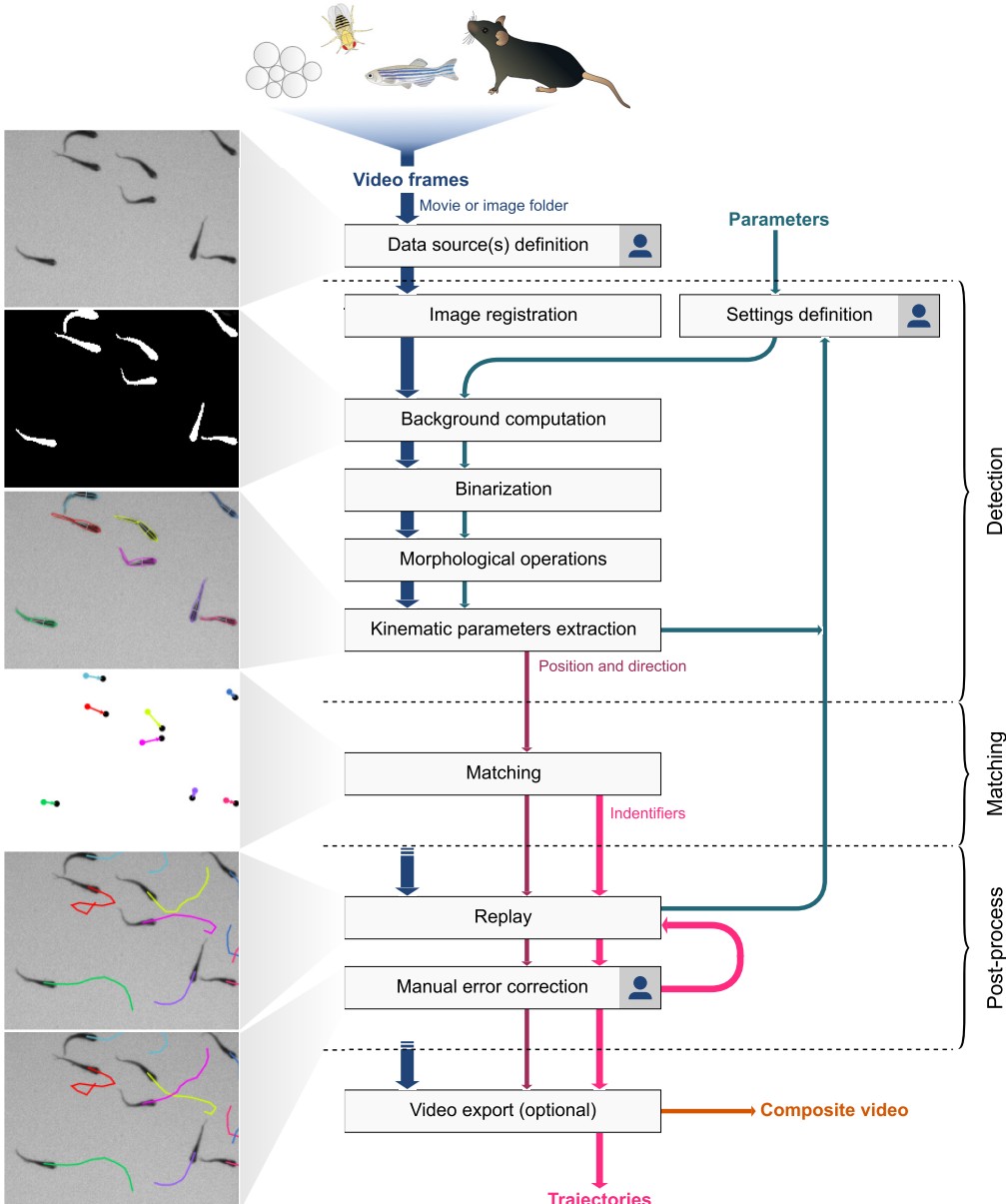

**Fig 1. FastTrack flow chart.** The workflow divides in three mains parts: detection, matching and post-process. The few steps that require user input are indicated by a 👤. Sample dataset: `ZFJ_001`.

three different approaches: phase correlation, enhanced correlation coefficient and feature-based. All methods are implemented in a pyramidal way, *i.e.* registration is first performed on downsampled images to correct for large drifts and then minute corrections are computed from full resolution images.

The phase correlation method detects translational drifts between two images by using the Fourier shift theorem in the frequency domain. This method is resilient to noise and artifacts but can misestimate large shifts. The enhanced correlation coefficient (ECC) registration method consists in maximizing the ECC function to find the best transformation between two images [28]. In FastTrack, the ECC registration is restraint to Euclidian motion (translation

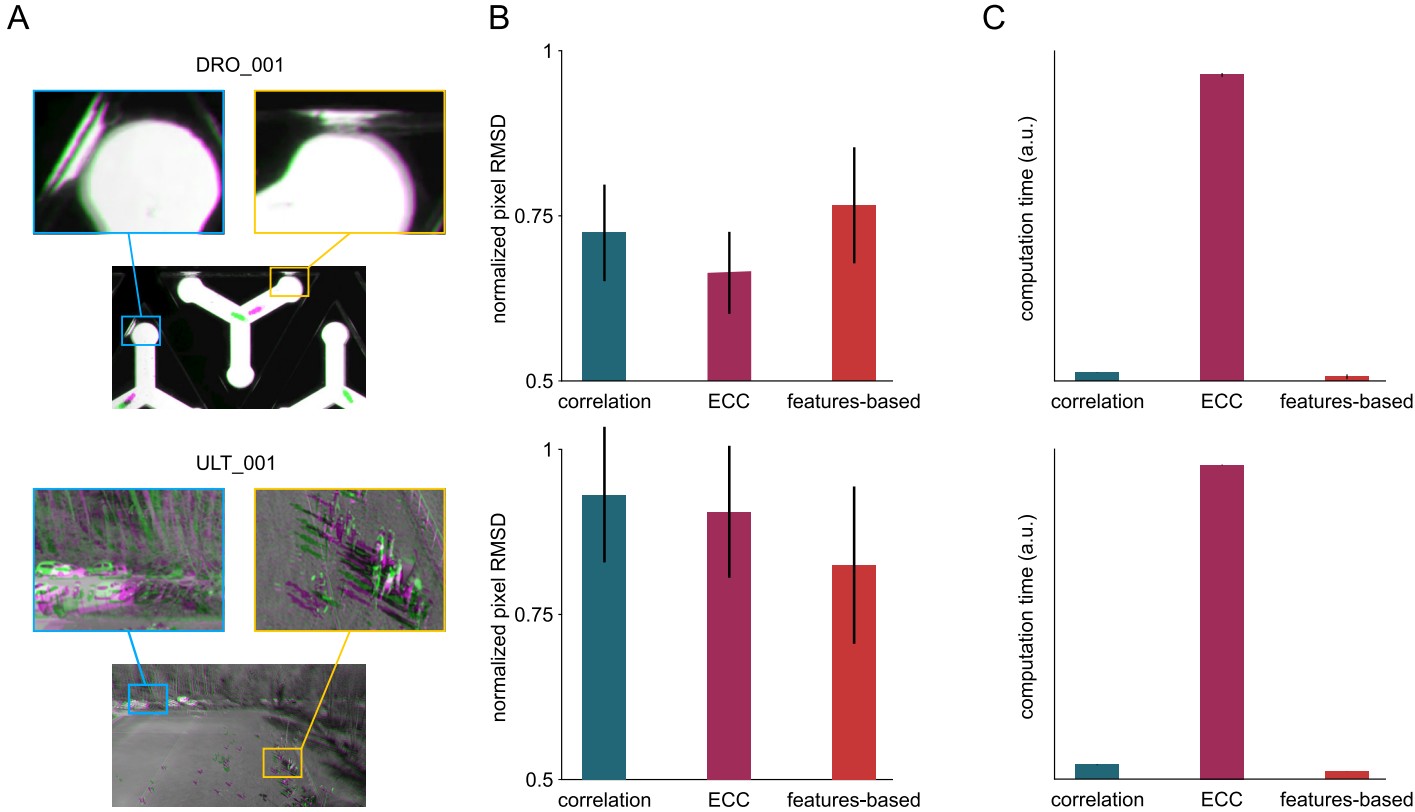

**Fig 2. Image registration.** Two recordings with severe drift are used for benchmarking (top: `DRO_001`, bottom: `ULT_001`). (**A**) Comparison of a frame (magenta) with the first frame (green) and magnification of details in the scene. (**B**) Root mean square deviation (RMSD) of pixel intensities after registration onto the first image, averaged over all time frames and normalized by the RMSD without registration, for three registration methods. Error bars: standard deviation across time frames. (**C**) Relative average computation time of the three registration methods, normalized by the total number of pixels in the movie (arbitrary units). Error bars: standard deviation across time frames. Raw data are available in S1 Data.

and rotation). This method has several assets, as it is invariant with respect to photometric distortion, performs well in noisy conditions and the solving time is linear, leading to an acceptable computation time with respect to other optimization algorithms [28] even though it is slower than the other two methods implemented here. Feature-based registration consists in finding key points and their descriptors in a pair of images and compute a homography. Then, the corresponding transformation is applied to all the pixels of one image to perform the registration. FastTrack uses the automatic ORB feature detector [29] to find approximately 500 key points in each image. The key points are matched pairwise between the two images using the Hamming distance. The homography is computed between the matching key points by using the Random Sample Consensus RANSAC [30] estimation method to minimize errors.

Fig 2 provides a rough comparison of the performance of the three methods. Using two recordings of the dataset, we benchmarked both the accuracy—with the root mean squared difference (RMSD) of pixel intensities between the reference and the corrected image—and the relative computation time. Choosing the right method to obtain the best accuracy depends on each movie's characteristics, but one can use the rule of thumb that if the objects to track occupy a large fraction of the total area then the best accuracy is more likely to be obtained by using ECC, and by using the features-based method otherwise. However, as shown in Fig 2C the ECC method is generally slower by an order of magnitude, so we recommend to use the features-based method in the general case, and *a fortiori* for long movies.

**Object detection.**   Object detection is performed by binarizing the difference between the frames and a background, and filtering the result. At each step, the display is live-updated as the user changes the parameters to provide a direct visual feedback.

In case the scene can be imaged without objects, or when specific computation are necessary, the background can be provided as a separate image file. Otherwise FastTrack can estimate the background by taking the minimum, maximum or average value of pixels on a subset of images taken at regular intervals. The user has then to specify the region of interest (ROI) in the images (default: full image), whether the background is lighter or darker than the objects to detect and provide a threshold to finalize the binarization.

A collection of standard operations is implemented to filter the binary images: morphological operations (erosion, dilatation, closing, opening, gradient, top hat, black hat and hit miss) with rectangle, cross-shaped or elliptical kernels, and a filter based on the area of the remaining connex shapes can be used to remove both small artifacts (shapes below the minimal area threshold) and overlapping objects (shapes above the maximal area threshold). Finally, the objects' contours are extracted from the remaining binary shapes by using the algorithm described in [31].

**Kinematic parameters extraction.**   Kinematic parameters are a collection of scalars that are extracted from the images in order to feed the matching algorithm. This step is at the core of any tracking procedure, and a large panel of parameters have been employed previously, ranging from basic measurements (*e.g.* position of the center of mass) to more complicated quantities aimed at identifying uniquely the object to track [13, 32].

Here, to improve speed the matching algorithm is based on quantities that are straightforward to obtain, namely the position of the center of mass, the angle of the binary object's main axis, the area and the perimeter. To extract the position and angle, FastTrack computes the equivalent ellipse of each object by using the second order image moments as in [33], but for an even faster implementation we derived it from the contours using the Green's formula [34]. The object's orientation is given by the ellipse's major axis and is only defined in the interval $[0; \pi[$. We determined the directions in the interval $[0; 2\pi[$ by projecting the pixels of each object on the major axis of the equivalent ellipse and calculating the skewness of the distribution of distances of these projected points to the center of mass. The sign of the skewness is a robust indicator of the asymmetry of the object along its principal axis.

However, for non-rigid objects this direction can significantly deviate from the direction of motion. For instance, swimming fish bend their body and the instantaneous direction of motion is closer to the head's direction than to the whole body's (*e.g.* MED_001, ZFJ_001, ZFA_001). We thus added as an option the method developed in [35–37] to decompose the objects in two ellipses—corresponding to the head and tail in the case of fish—and let the user determine which ellipse is more representative of the direction of motion (S2 Fig).

## Matching

In this step, objects on different frames are paired up based on the similarity of their kinematic parameters. FastTrack uses 4 kinematic parameters (position, direction, area and perimeter) in a method inspired by [38], which relies on the fact that objects usually change very little in position or direction between successive frames, as compared to their relative distances and angular difference.

For each pair of objects $(i, j)$ belonging to distinct time frames, and for each kinematic parameter, FastTrack computes a cost which is the product of a *soft* and a *hard* term (see S3 Fig). This terminology is brought from statistical physics, where particles can have soft, long-ranged interactions or hard, binary contacts. For instance, with position as kinematic

parameter the hard term $h_r^{i,j}$ is set to 1 when the distance between $i$ and $j$ is smaller than a given threshold $h_r$ and set to $+\infty$ otherwise. The soft term $s_r^{i,j}$ is simply computed as the distance $dr_{i,j}$ normalized by a factor $s_r$. Altogether, the complete cost function for a pair $(i, j)$ writes:

$$c_{i,j} = h_r^{i,j} \frac{dr_{i,j}}{s_r} + h_\alpha^{i,j} \frac{d\alpha_{i,j}}{s_\alpha} + h_A^{i,j} \frac{dA_{i,j}}{s_A} + h_p^{i,j} \frac{dp_{i,j}}{s_p} \qquad (1)$$

where $dr_{i,j}$ is the distance between $i$ and $j$, $d\alpha_{i,j}$ is the absolute angular difference between the directions of $i$ and $j$ defined in the interval $[0; \pi]$ and $dA_{i,j}$ and $dp_{i,j}$ are the absolute difference of area and perimeter. If any of the hard thresholds is passed, then $c_{i,j} = +\infty$ and the association is impossible. The soft factors $s_r$, $s_\alpha$, $s_A$ and $s_p$ are normalization coefficients which represent the typical changes that an object undergoes between the two frames. In case one would like to discard a kinematic parameter from the computation, both the hard threshold and the soft factor have to be set to $+\infty$.

The cost matrix can be rectangular if the number of objects is not constant, typically when there are occlusions, object loss during the detection phase or entries and exits at the boundaries of the ROI. Finding the best matching amounts to define the set of pairs that minimizes the sum of costs for all retained pairs. This problem, sometimes called the *rectangular assignment problem*, falls into the category of linear assignment problems [39] and can be exactly solved using the Kuhn-Munkres algorithm [40], also called the *Hungarian algorithm*. FastTrack uses a fast C++ implementation [41] of this algorithm to perform the matching automatically.

The Kuhn-Munkres algorithm operates only between two time frames, so in strict implementations when an object disappears the trajectory stops, and if it reappears later on a new trajectory is created. In order to deal with brief object disappearances we introduce a temporal hard parameter $h_t$, which is the maximal acceptable time during which an object can be lost. In practice, all the objects that have not been assigned in the few previous frames (below $h_t$) are also integrated in the cost matrix for the matching between $t$ and $t + 1$. They are treated as pseudo-occurences at time $t$, though the resulting trajectories are generated such that they appear at the correct time frame, which is before $t$. This allows the algorithm to have a kind of "short-term memory" and manage short disappearances due to detection issues or occlusions for instance.

In the graphical interface of FastTrack, the user can set up the tracking parameters and preview the tracked trajectories on a selected chunk of the image stack.

## Post-processing

**Output.**   FastTrack delivers the tracking result in a single, large text file with one row per object per frame. This format is convenient since it can be parsed for subsequent analysis with many external tools. The array contains 23 columns corresponding to the features tracked; the main features are the position and direction of the object (`xBody`, `yBody`, `tBody`), the object's id (`id`) and the frame index (`imageNumber`). The other features are self-explanatory and provided to the user for optional subsequent analysis.

Additionnally, a movie with the tracking overlaid can be created from the *Replay* panel and saved in `AVI` format.

**Manual post-processing.**   The *Replay* panel can be used to manually correct for errors in the trajectories. The output text file can be loaded at any time and the display overlays tracking information like the objects' indices or short-term anterior trajectories onto the original frames. Graphical controls as well as a set of keyboard shortcuts provide an efficient

framework for spotting errors, remove fragments of trajectories and switch indices whenever necessary.

## Results

### Processing of the dataset

Being able to track objects in virtually any movie is an overwhelming challenge, and achieving this with one single software is, to date, utopic. This is primarily due to the versatility of the imaging conditions which multiply the number of approaches that are needed to correctly perform the detection phase. Two main pitfals can be discerned: the variations due to illumination (*e.g.* reflections as in GRA_001, shadows as in SOC_001 and TRA_001) and the partial overlaps between objects (*e.g.* HXB_001, ZFA_001).

Still, many experimental setups in academia and production lines in industry are designed to mitigate these issues and it is common to have a uniform, diffuse and constant illumination, and either a compartimentation, strict two-dimensional confinement or low density of objects to avoid overlaps. In the TD$^2$ dataset, approximately half of the movies (23, see S1 Table) had sufficiently good illumination conditions to be tracked directly with FastTrack. The other movies required an additional pre-processing step, that was performed upstream with custom Matlab scripts. Severe and frequent overlaps impeded the processing of two movies (HXB_001 and ZFL_001), that were discarded for the remaining of the analysis.

Then, the workflow performed robustly and we could track the objects in the remaining 39 movies. The Kuhn-Munkres algorithm performs in $O(n^3)$ polynomial time [42] so the matching phase is generally fast; on a modern workstation we observed processing peaks at 500 frames/s and it took at most a few minutes with up to more than 2,000 objects in the field of view over 1,000 frames (GRA_001). We then used the post-processing tools of FastTrack to manually correct for all remaining errors, and we could achieve a perfect tracking—within the margin of a few human errors that we may have missed—for all the dataset (S1 Video) in a reasonable time. In the following, we refer to these trajectories as the *ground truth* trajectories.

### Classification of the dataset based on incursions

To characterize and classify the different movies of the dataset, we introduce the notion of "incursions". Incursions happen every time an object travels sufficiently between to time points to exit its own Voronoï cell, defined at the initial time, thus entering one of it's neighbor's cell. With this definition the Voronoï cells are static boundaries defined by the initial frame, and the likelihood of an incursion highly depends on the timescale $\tau$ over which the displacement is observed (Fig 3A), with $\tau = 1$ corresponding to the time between two successive frames. On a global scale, the amount of incursions depends on the distribution of displacements but also on the density $d$ (defined as the number of objects per unit area), the complex statistical properties of the geometry of Voronoï cells and the degree of motion alignment.

To account for the density, we use the dimensionless reduced displacement $\rho = r\sqrt{d}$. A value of $\rho = 1$ represents the typical distance between two objects, while $\rho = 1/2$ is the typical distance for an object to travel before entering a neighbor's Voronoï cell. Then, for a given displacement $\rho$ we compute the *geometric probability of incursion* $p_{inc}(\rho)$ by determining the proportion of angles for which incursions occur, on average over a representatively large set of Voronoï cells. As illustrated in Fig 3B, $p_{inc}(\rho)$ always has a sigmoid shape with an inflection point close to $\rho = 1/2$. The precise shape of this function is sensitive to the compacity of the objects in the scene: if they are sparsely distributed (*e.g.* PAR_001, ROT_001) then the

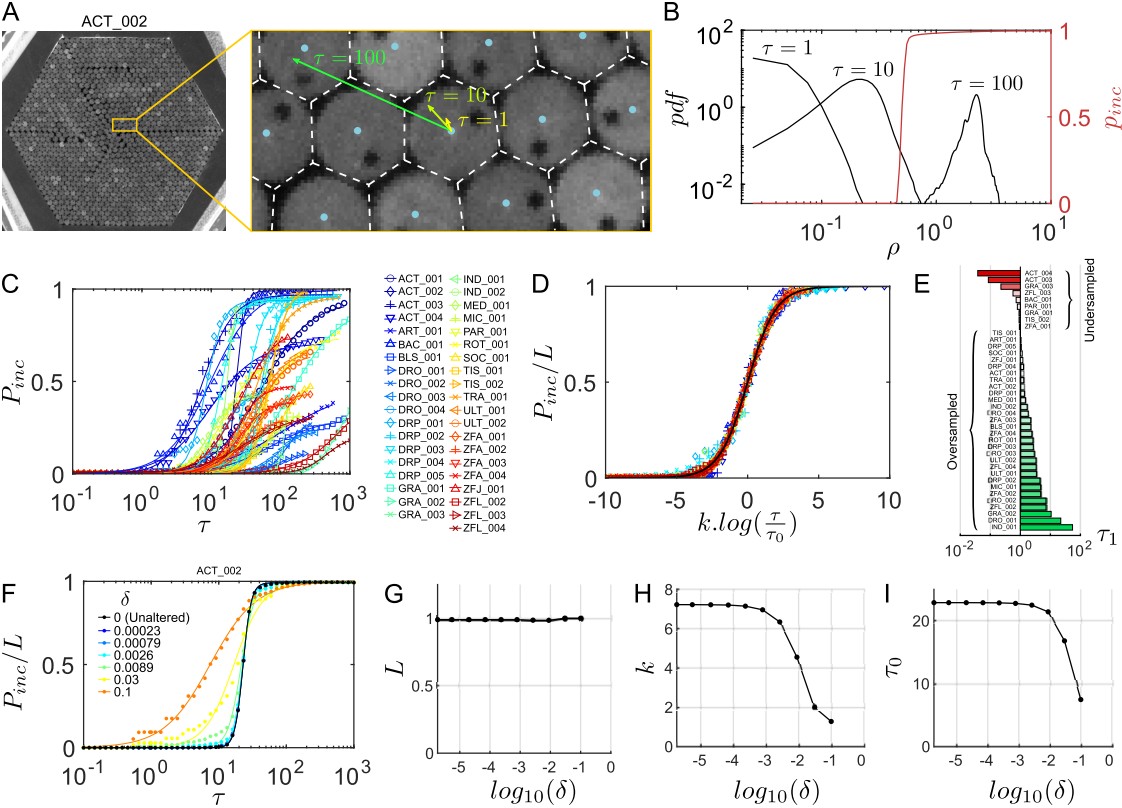

**Fig 3. Characterization of the TD² dataset.** (**A**) Illustration of the dynamics at various timescales in `ACT_002`. The Voronoï cells (dashed white) and the displacements of a particle at $\tau$ = 1, 10 and 100 are overlaid. (**B**) Geometric probability of incursion $p_{inc}$ (red) and distribution of the reduced displacement $\rho$ at three different timescales $\tau$ (black) in `ACT_002`. The probability of incursion $P_{inc}$ is the intersection of the areas under the two curves. (**C**) $P_{inc}$ as a function of $\tau$ for the whole dataset (symbols). The solid lines are fits with a logistic function (see text). (**D**) Scaling of the reduced quantities $P_{inc}/L$ as a function of $k.log(\frac{\tau}{\tau_0})$ on the standard logistic sigmoid function (solid black). (**E**) Classification of the movies in the dataset by increasing values of $\tau_1$ as defined by Eq (4), with fitting parameters determined over a logarithmic scale for $P_{inc}$. Movies with $\tau_1 < 1$ are undersampled while movies with $\tau_1 > 1$ are oversampled. (**F**) Comparison of $P_{inc}(\tau)$ for different levels of degradation $\delta$ (symbols) and corresponding logistic fits (solid curves) in `ACT_002`. (**G-I**) Evolutions of the fitting parameters $L$, $k$ and $\tau_0$ as a function of the degration $\delta$ in `ACT_002`.

Voronoï cells are highly heteregeneous and $p_{inc}$ growns slowly, while for dense packings forming an hexagonal pattern (*e.g.* `ACT_002`, `DRP_001`) the cells are stereotyped and $p_{inc}$ increases steeply. The asymptotic value of $p_{inc}$ for $\rho \gg 1$ may not be 1 for systems with reflective walls and a low number of objects, as show in S4 Fig.

Assuming that the dynamics is uncorrelated with the geometric properties of the Voronoï cells, the probability of incursion writes:

$$P_{inc} = \int_0^\infty R(\rho)p_{inc}(\rho)d\rho \qquad (2)$$

where $R(\rho)$ is the distribution of $\rho$ at the timescale $\tau$. The distribution $R(\rho)$ is shown in Fig 3B for three values of $\tau$, and a graphical way of calculating $P_{inc}$ is to take the intersection of the areas under $R(\rho)$ and $p_{inc}(\rho)$. In the regime where $R(\rho)$ and $p_{inc}(\rho)$ are well-separated, the resulting value of $P_{inc}$ are low but also highly sensitive to the amount of swaps in the tracking; indeed, the swaps create a bump in $R$ at values of $\rho$ close to one that can artificially increase $P_{inc}$ of orders of magnitude. So, unless the ground truth trajectories are accessible, in most

cases the single value of $P_{inc}$ at $\tau = 1$ cannot be used as an *ad hoc* measure for the trackability of a movie.

A timescale-varying analysis allowed us to extract more robust quantifiers. As $p_{inc}(\rho)$ does not depend on $\tau$ and $R(\rho)$ is shifted to the high values of $\rho$ when $\tau$ increases, one can easily expect that $P_{inc}(\tau)$ has a sigmoid-like shape. We thus computed $P_{inc}$ for various values of $\tau$: for $\tau > 1$ we tooks integer values (*i.e.* keep one frame every $\tau$) while for $\tau < 1$ we linearly interpolated the displacements (*i.e.* multiplied $\rho$ by $\tau$). We represented the results in Fig 3C for the 39 movies that could be tracked in the dataset. Strikingly, all $P_{inc}$ followed a logistic curve when $\tau$ is log-scaled so we used fits of the form:

$$P_{inc} = \frac{L}{1 + e^{-k(log(\tau) - x_0)}} \tag{3}$$

and, noting $\tau_0 = e^{x_0}$, the fitting function can be rewritten as:

$$P_{inc} = \frac{L}{1 + \left(\frac{\tau_0}{\tau}\right)^k} \tag{4}$$

The fits are shown in Fig 3C, and are valid for all the movies in the dataset. To make all data collapse on a single master curve, we show in Fig 3D that $P_{inc}/L$ plotted as a function of $k.log(\frac{\tau}{\tau_0})$ follows the standard logistic sigmoid function $f(x) = \frac{1}{1 + e^{-x}}$.

An interesting outcome of this approach is the ability to determine the framerate at which experiments should be performed. It is indeed a recurrent experimental question, as high temporal resolution is preferable to reduce the number of incursions and ease the tracking, but may not always be accessible (*e.g.* limited sensor rate, intense illumination required as the exposure time drops) and generates large amounts of images to store and process. We computed $\tau_1$, the timescale at which $P_{inc}$ reaches the inverse of the total number of objects on all frames $N_{obj}$, *i.e.* the probability of a single incursion in the whole movie. As $\tau_1$ defines the onset of incursions and the possibility of swaps in the tracking procedure, it can be used as an indicator of the sampling quality of each movie. Movies with $\tau_1 < 1$ already have incursions at the current framerate and are thus *undersampled*, whereas for movies with $\tau_1 > 1$ the current framerate can be degraded without triggering incursions, they are *oversampled*. In addition, $\tau_1$ is directly the resampling factor that one should use to have the minimal movie size without generating incursions. Using Eq (4), it reads:

$$\tau_1 = \tau_0(LN_{obj} - 1)^{-\frac{1}{k}} \tag{5}$$

We ordered the values of $\tau_1$ in Fig 3E, and it appears that three quarters (30) of the movies are oversampled; one should not expect any difficulty in the tracking of these movies with respect to incursions. On the other hand, the 9 undersampled recordings were already known to be difficult to track; three of them (`ACT_003`, `ACT_004` and `GRA_003`) have required specific algorithms for analysis [19, 20, 24] and two (`BAC_001`, `ZFA_001`) required dedicated softwares [13, 43, 44].

Then, we tested to what extent this characterization is robust to swaps in the trajectories. Starting from the ground truth trajectories of `ACT_002`, we degraded the trajectories by introducing random swaps between neighboring objects. This process is controlled by a degradation rate $\delta$, which is the number of artificial swaps divided by the total number of objects on all frames. Such a degradation affects the small timescales more severely, so the multi-scale approach takes on its full interest: as depicted in Fig 3F to 3I, the fits of $P_{inc}(\tau)$ are insensitive to degradation up to a remarkably high level of $\delta \simeq 10^{-3}$. This means that even a poor-quality

tracking can be used as an input for this method: as long as the distribution of displacements is only marginally affected, the output remains unchanged.

## Automatic tracking parameters

In this section we detail the procedure used by FastTrack to determine automatically the soft normalization factors ($s_r$, $s_\alpha$, $s_A$ and $s_p$) which, according to Eq (1), allow to compare terms of very different nature and amplitude into a single cost function. It is therefore intuitive to use the standard deviation of the increments of each kinematic parameter. However, as one needs some trajectories in order to estimate the standard deviations, we set up an iterative, rapidly-converging algorithm.

Let us use `ZFJ_001`, a movie that is slightly oversampled but with many occlusions and objects of different sizes to illustrate the details of the algorithm. For the sake of simplicity let us use here only the position, angle and area as kinematic parameters—it is straightforward to add the perimeter to the method but there is no gain to expect as the objects shapes are very similar. A snapshot of this movie is shown in Fig 4A.

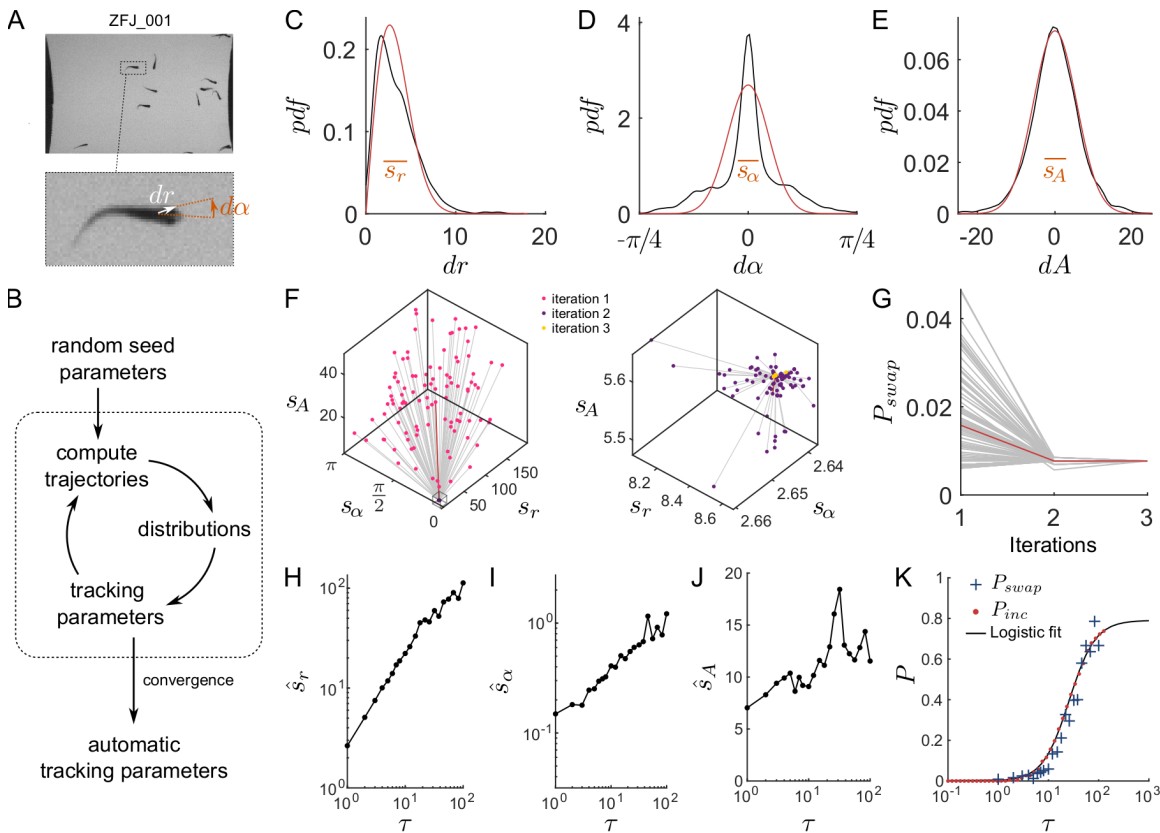

**Fig 4. Automatic tracking parameters.** (**A**) Snapshot and blow-up of `ZFJ_001`, with definition of $\vec{dr}$ and $d\alpha$ (**B**) Scheme of the algorithm for determining the tracking parameters automatically. (**C-E**) Distribution of displacements $dr$ (in pixels), angular differences $d\alpha$ (in radians) and area differences $dA$ (in pixels) when the default parameters of the software are used on `ZFJ_001`, for $\tau = 1$ (black). The corresponding $\chi$ and Gaussian fits are displayed in red. Orange bars: resulting soft parameters. (**F**) Evolution of $s_r$, $s_\alpha$ and $s_A$ with algorithm iterations for `ZFJ_001`. Left: iterations 1 and 2; right: iterations 2 and 3. A hundred runs with random initial values are shown, the run with the software default parameters is highlighted in red. (**G**) Evolution of $P_{swap}$ with algorithm iterations, same runs. (**H-J**) Evolution of the converged parameters $\hat{s}_r$, $\hat{s}_\alpha$ and $\hat{s}_A$ as a function of the timescale $\tau$ for `ZFJ_001`. (**K**) Comparison between $P_{swap}$ (blue crosses) obtained with the converged parameters and $P_{inc}$ (red dots) for `ZFJ_001`. The solid black line is the logistic fit of $P_{inc}$.

In order to evaluate the distributions of $dr$, $d\alpha$ and $dA$, we start by tracking the movie with the default parameters of the software. The resulting distributions are shown in Fig 4C to 4E. Then, for kinematic parameters whose differential values can be positive or negative the distribution is fitted by a Gaussian function and the soft parameter is set at the standard deviation. For instance with the angular difference $d\alpha$ the fit reads:

$$f(d\alpha) = \frac{1}{s_\alpha \sqrt{2\pi}} \, e^{-\frac{d\alpha^2}{2s_\alpha^2}} \qquad (6)$$

and $s_\alpha$ (orange bar in Fig 4D) is stored as the soft parameter to use during the next iteration. The computation of the soft parameter for displacement $s_r$ is different since distances can only be positive. Assuming that the displacements along the $x$ and $y$ axes follow two independent Gaussian processes, the resulting displacement follows a $\chi$ distribution with 2 degrees of freedom and the fit reads:

$$f(dr) = \frac{dr}{s_r^2} \, e^{-\frac{dr^2}{2s_r^2}} \qquad (7)$$

where $s_r$ (orange bar in Fig 4C) is stored as the soft parameter to use for the next iteration.

Once all tracking parameters have been derived fom the distributions, the software recomputes new trajectories and updates the distributions. This iterative process, depicted in Fig 4B, is run until the tracking parameters converge. In practice, the convergence is very fast regardless of the initial position in the parameters space: we drawn 100 sets of seed parameters from uniform distributions spanning large intervals and convergence has been attained in a very few iterations for all parameters (Fig 4F). The convergence criterion implemented in the software is that all parameters should vary that less than $10^{-3}$ of the corresponding soft parameter.

We then characterized the resulting trackings by computing the amount of swaps with respect to the ground truth, captured by the probability of swaps:

$$P_{swap} = \frac{N_{swap}}{N_{obj} - n_{ap}} \qquad (8)$$

with $N_{swap}$ being the total number of swaps, $N_{obj}$ the total number of objects on all frames and $n_{ap}$ the number of times a new object appears. If the number of objects is constant and noted $n$, then $n_{ap} = n$ and $N_{obj} = nT$, with $T$ the number of frames in the recording, such that $P_{swap}$ simply reads:

$$P_{swap} = \frac{N_{swap}}{n(T-1)} \qquad (9)$$

$P_{swap}$ also converges very fast, to a value that is nearly-optimal: for 77% of the parameter sets $P_{swap}$ is decreased or remain equal, with an average drop of 0.0119 (155% of the converged value), while for 23% of the parameter sets $P_{swap}$ is increased with an average rise of 0.0011 (14% of the converged value). The expected difference is thus −0.0090 (116% of the converged value) for this movie. The automatic parameters are therefore a very good starting point in the general case, over which the end user can fine-tune the weights given to the kinetic parameters to take into account the specificities of each movie.

Then, we computed the converged soft parameters $\hat{s}_r$, $\hat{s}_\alpha$ and $\hat{s}_A$ for several sampling rates of $\tau > 1$ (Fig 4H to 4J). We then used these parameters to track the ZFJ_001 movie at different $\tau$ and compute $P_{swap}$. A comparison between $P_{swap}$ and $P_{inc}$ as a function of $\tau$ is shown in Fig 4K. This comparison illustrates that $P_{swap}$ is a noisier measurement of a movie's trackability

than $P_{inc}$ and confirms that the iterative algorithm produces trajectories with an amount of errors that is close to the statistical limit.

## Parameter optimization

One may also want to determine the tracking parameters that are really optimal with respect to $P_{swap}$. In this case, provided that the ground truth is known (*e.g.* by a careful manual post-processing) for at least one movie, it is possible to leverage the speed of FastTrack to explore the parameters space and minimize $P_{swap}$; then the optimized parameters can be used to track other similar movies with a minimal error rate. The workflow of the method is depicted in S5(A) Fig. As the exploration of the whole parameters space requires to perform at least thousands of trackings, such an approach is only made possible by the command-line interface (CLI, see S2 Table) and the speed of execution of FastTrack.

Let us first apply this approach to gain insight into the influence of $h_r$, the maximal distance allowed for an object to travel before it is considered lost. S5(B) Fig displays how $P_{swap}$ evolves as a function of $h_r$ for three recordings in the dataset. For low values of $h_r$, $P_{swap}$ is essentially imposed by the distribution of the objects' displacements, since a high number of errors are generated when the objects are not allowed to move sufficiently. For higher values of $h_r$, the distribution of the distances to the neighbors—as defined by the Voronoï tesselation—starts to influence $P_{swap}$ as the algorithm becomes sensitive to incursions. It can also be more easily fooled by entries and exits at the boundaries of the region of interest when the number of objects in the scene varies.

In between, for most recordings there is a gap yielding the minimal probability of error; this is particularly true when the objects are densely packed, since the distribution of distances to neighbors is sharper, like for `DRP_001` where $P_{swap}$ drops to zero on a range of $h_r$. The acquisition framerate also has an important role here: with highly-resolved movies the distribution of displacements is shifted to the left, leading to a clear separation and low values of $P_{swap}$, while for poorly-resolved movies like `ZFJ_001` the two distributions overlap and $P_{swap}$ is always bound to high values.

Similar analysis can be performed on the other tracking parameters. S5(C) Fig represents $P_{swap}$ as a function of both hard parameters $h_r$ and $h_t$ for `PAR_001`, and a thin optimal segment appears. S5(D) Fig represents $P_{swap}$ as a function of the two soft parameters $s_r$ and $s_\alpha$, and an optimal ratio lies at $s_r / s_\alpha \simeq 0.63$. Altogether, a set of optimal parameters can be derived and used for the processing of similar recordings.

## Comparison with other softwares

To assess the performance of FastTrack, we ran a benchmark with two state-of-the art tracking softwares that have been applied to various types of two-dimensional data: idtracker.ai [45] and ToxTrac [46]. First, it is worth mentionning that these softwares are much more difficult to install than FastTrack, and that they have strong intrinsic limitations as compared to Fast-Track: both require a good framerate and image quality, with sufficient contrast and number of pixels per object, and a constant number of objects in the scene.

The benchmark was thus performed on a dataset constituted of a selection of videos that were provided with each software and some movies of the TD$^2$ dataset that meet these requirements. `idtrackerai_video_example` and `100Zebra` (S4 Video) are available on the idtracker.ai website (https://idtrackerai.readthedocs.io/en/latest/data.html). `Guppy2`, `Waterlouse5`, and `Wingedant` on the ToxTrac SourceForge (https://sourceforge.net/projects/toxtrac/files/ScientificReports/). Movies that were provided in image sequence format were converted losslessly to video format with FFmpeg since idtracker.ai and ToxTrac were

not able to process directly image sequences. `DRO_002` and `ACT_002` were preprocessed with a custom Matlab script to detect the objects before using the softwares. In addition, only the first 100 images of `DRO_002` were used to reduce the computing time.

The benchmark between idtracker.ai and FastTrack was performed on a workstation with an Intel i7-6900K (16 cores), 4.0 GHz CPU, an NVIDIA GeForce GTX 1060 with 6GB or RAM GPU, 32GB of RAM, and a NVMe SSD of 250GB running Arch Linux. The parameters were set by trials and errors inside the graphical user interface of the two softwares. The tracking duration was recorded using the command line interface available for the two software. The average tracking duration and the standard deviation were averaged over 5 runs except for `DRO_002` (2 runs) and `ACT_002` (1 run) due to the very long processing time. Idtracker. ai was evaluated with and without GPU capability except for `100Zebra`, `DRO_002`, and `ACT_002` due to the very long processing time.

The benchmark between ToxTrac and FastTrack was performed on a computer with an Intel i7-8565U (4 Cores), 1.99 GHz CPU, 16 GB of RAM, and a NVMe SSD of 1 TB running Windows10. The parameters were set by trials and errors in the graphical user interface. The average tracking duration and the standard deviation were averaged over 5 runs using the built-in timer feature implemented in each software.

The accuracy was evaluated manually using the built-in review feature implemented in each software. The number of swaps and the number of non-detected objects were counted in each movie. Occlusion events were ignored in this counting. The accuracy was computed as:

$$A = \frac{n_{obj} * n_{img} - (2 * N_{swap} + N_{undetected})}{n_{obj} * n_{img}} \tag{10}$$

with $N_{swap}$ the number of swaps, $N_{undetected}$ the number of non-detected objects, $n_{obj}$ the number of objects and $n_{img}$ the number of images. For `100Zebra`, the accuracy was computed only over the 200 first images.

All the results are presented in Fig 5. In terms of speed FastTrack is orders of magnitude faster than idtracker.ai and significantly faster than ToxTrac on all tested videos. In terms of accuracy, all softwares performed extremely well, except idtracker.ai on `ZFJ_001`. FastTrack had a perfect accuracy on 8 movies out of 11 and always had an accuracy above 99%. In order to correct for the few errors, the ergonomic post-processing interface of FastTrack can be used to reach a perfect tracking within a few more minutes. Altogether, FastTrack offers many assets as compared to idtracker.ai and ToxTrac: it is more versatile and the total time spent to track a movie is globally lower—in some cases by orders of magnitude—without sacrificing the tracking accuracy.

## Availability and future directions

The source code of FastTrack is licensed under the GNU GPLv3 and can be downloaded at the following Github repository: https://github.com/FastTrackOrg/FastTrack. Developers implementing algorithmic improvements or new features are encouraged to submit their code *via* Github's pull request mechanism for inclusion in a forthcoming release.

Pre-built binaries (AppImages) for Windows, Mac and all versions linux can be downloaded on the FastTrack website (http://www.fasttrack.sh). We also provide a PPA for Ubuntu 18.04 that ensure proper system integration. The Windows and PPA versions incorporate an automatic update manager such that users always stay up-to-date with the latest stable release. For the Mac and Linux AppImages, updates have to be performed manually by downloading the latest version.

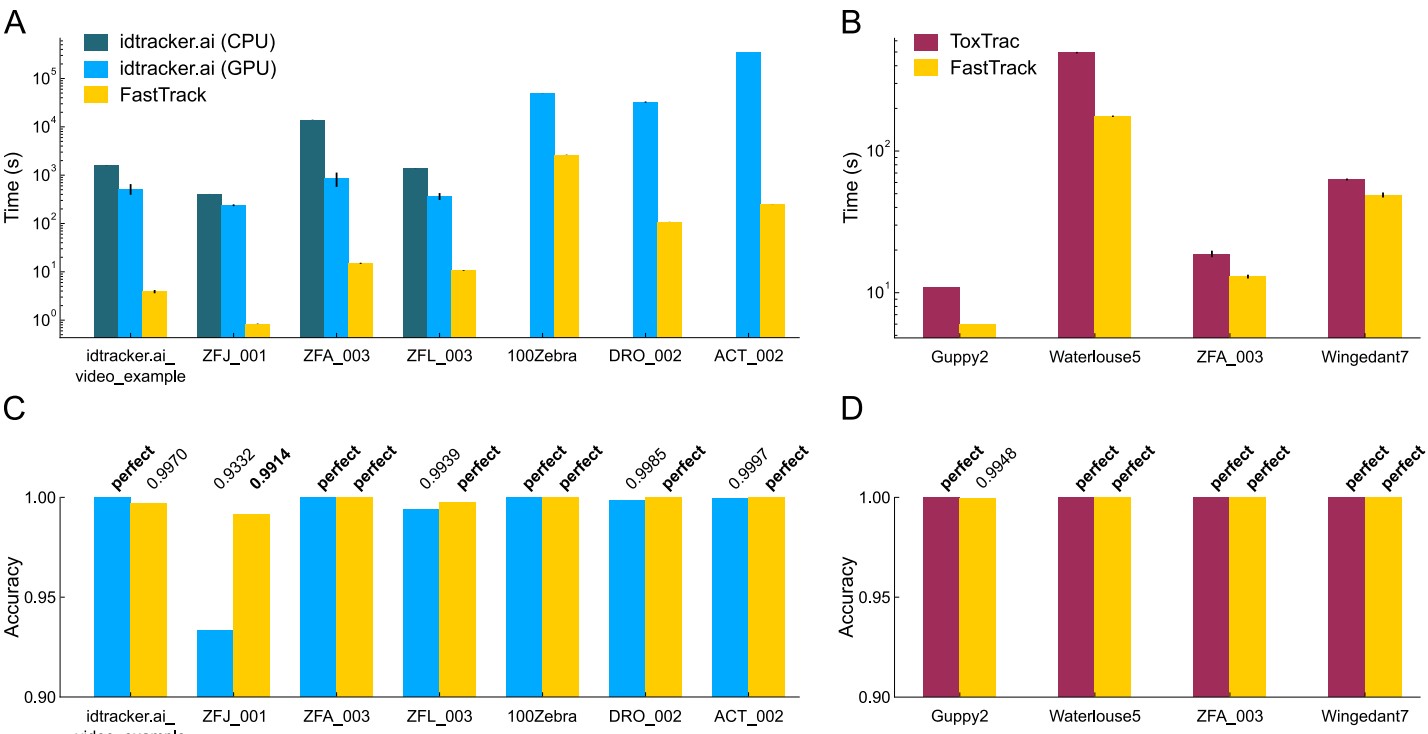

**Fig 5. Benchmark of FastTrack, idtracker.ai and ToxTrac.** (**A-B**) Comparison of the computation time for the tracking of various movies with the same workstation. Whenever possible, CPU and GPU variants of idtracker.ai have been run. Only the first 100 images of `DRO_002` have been used. (**C-D**) Accuracies of the resulting trackings. "perfect" means an accuracy of exactly 1. The trajectories computed by the CPU and GPU variants of idtracker.ai being rigourously similar, we only show the results for the GPU. For `100Zebra`, the accuracy was computed by taking into account only the first 200 images.

User's and developer's documentation are available on the FastTrack website. A video tutorial for beginners illustrating the complete workflow is available in S2 Video. We also provide an additional video presenting the post-processing panel in detail (S3 Video).

The FastTrack algorithm only uses past and current information and is relatively fast, so it is in principle amenable to live tracking. Currenty the software doesn't natively integrate a live tracking feature, but it can be integrated inside any C++ project with minimal programming skills as explained in the developer's documentation (http://www.fasttrack.sh/API/index. html).

In the future, we will continue the effort to make the TD$^2$ dataset grow, and greatly encourage new submissions from all fields of science. It is a useful basis for the development of general-purpose tracking softwares and a convenient material for benchmarking.

Though the software has been designed to process a broad range of systems, there is still room for improvement both at the detection and matching levels to make it more universal. For the detection phase, the need for an upstream custom preprocessing step could be greatly reduced for many movies by the implementation of tools for shadow removal [47] and adaptative thresholds [48]. The separation of the RGB channels for colored images could also be useful in some cases. Then, for the matching phase we plan to allow for the integration of more kinematic parameters in the computation of the cost matrix. Among the possible improvements we can mention basic shape descriptors like eccentricity or the object's velocity. In the latter case, it may be useful to modulate the hard threshold for displacements $h_r$ by the previous values of speed. We also plan to integrate the average intensity level, which could be useful for tracking fluorescent objects with different levels of expression.

Finally, in order to overcome the problem of losing the objects when they are clipped by a boundary or overlapping, we are currently working on an algorithm capable of resolving truncated and overlapping shapes based on a model learned in an unsupervised manner. Ultimately, this will make FastTrack a very complete tool for general tracking purpose.

## Supporting information

**S1 Table. Two-dimensional tracking dataset.** Description and credentials of the data that have been used for testing the FastTrack software. All movies in the dataset can be downloaded at http://data.ljp.upmc.fr/datasets/TD2.
(PDF)

**S2 Table. User-defined parameters.** List of the parameters that can be set in FastTrack for image processing and automatic tracking, as of version 5.1.7.
(PDF)

**S1 Fig. Thumbnails of the TD$^2$ dataset.**
(TIF)

**S2 Fig. Definition of the kinematic parameters.** The raw image is binarized and the binary shape is described by one or two ellipses. The position (center of mass) and direction (see text) of the chosen ellipse are used as an input for the matching phase.
(TIF)

**S3 Fig. General workflow of cost-based tracking.** Depending on the system and recording conditions, many kinematic parameters can be employed to define the cost matrix. For each parameter a soft (normalized) and a hard (binarized) terms can be combined and summed to form the General Cost Matrix. An assignment algorithm is then used to produce the trajectories.
(TIF)

**S4 Fig. Effect of confinment on $p_{inc}$.** The geometric probability of incursion $p_{inc}$ is computed for a system composed of $n$ punctual objects uniformly distributed at random in a square of size 1, as a function of the reduced displacement $\rho$ for various values of $n$ (plain). $p_{inc}(\rho)$ is also shown for a system without walls (dashed); in this case the curve is independent of the density $d$.
(TIF)

**S5 Fig. Optimization of tracking parameters based on $P_{swap}$.** (**A**) Scheme of the optimization workflow: on top of the detection/matching/post-process flow chart, the ground truth is used to compute $P_{swap}$ and create a feedback loop on the tracking parameters. (**B**) $P_{swap}$ (black) as a function of the maximal distance parameter $h_r$ (in pixels) for three typical recordings. Vertical lines for `DRP_001` indicate that $P_{swap}$ drops to 0. The distributions of displacements between successive frames (blue) and of distances to the neighbors (orange) are also shown for comparison. (**C**) $P_{swap}$ as a function of the maximal distance parameter $h_r$ (in pixels) and the maximal disappearance time $h_t$ (in frames) for `PAR_001`. Soft parameters are set to $s_r = 95$ and $s_\alpha = 60$. (**D**) $P_{swap}$ as a function of the normalization distance parameter $s_r$ (in pixels) and the normalization angle $s_\alpha$ (in degrees) for `PAR_001`. Hard parameters are set to $h_r = 210$ and $h_t = 90$.
(TIF)

**S1 Video. Result of the tracking with FastTrack.** Compilation of short sequences extracted from the 39 recordings of the TD$^2$ dataset that could be tracked with FastTrack. The

trajectories are overlaid over the original frames.
(MP4)

**S2 Video. General usage of the FastTrack software.**
(MP4)

**S3 Video. Details of the post-processing panel.**
(MP4)

**S4 Video. Tracking of the 100Zebra movie by FastTrack.**
(MP4)

**S1 Data. Raw data.**
(XLSX)

## Acknowledgments

We warmly thank all contributors to the TD$^2$ dataset. We thank Guillaume Le Goc, Julie Lafaye and Nicolas Escoubet for useful comments and feedback on the software.

## Author Contributions

**Conceptualization:** Raphaël Candelier.

**Data curation:** Benjamin Gallois, Raphaël Candelier.

**Formal analysis:** Raphaël Candelier.

**Funding acquisition:** Raphaël Candelier.

**Investigation:** Benjamin Gallois, Raphaël Candelier.

**Methodology:** Raphaël Candelier.

**Project administration:** Raphaël Candelier.

**Software:** Benjamin Gallois.

**Supervision:** Raphaël Candelier.

**Validation:** Benjamin Gallois, Raphaël Candelier.

**Visualization:** Benjamin Gallois.

**Writing – original draft:** Benjamin Gallois, Raphaël Candelier.

**Writing – review & editing:** Benjamin Gallois, Raphaël Candelier.

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
