## [Decision Letter · Decision Letter 0]

30 Jul 2020

Dear Dr Candelier,

Thank you very much for submitting your manuscript "FastTrack: an open-source software for tracking varying numbers of deformable objects" for consideration at PLOS Computational Biology.

As with all papers reviewed by the journal, your manuscript was reviewed by members of the editorial board and by several independent reviewers. In light of the reviews (below this email), we would like to invite the resubmission of a significantly-revised version that takes into account the reviewers' comments.

We cannot make any decision about publication until we have seen the revised manuscript and your response to the reviewers' comments. Your revised manuscript is also likely to be sent to reviewers for further evaluation.

Sincerely,

Manja Marz

Software Editor

PLOS Computational Biology

Manja Marz

Software Editor

PLOS Computational Biology

Reviewer's Responses to Questions

**Comments to the Authors:**

Reviewer #1: The review is uploaded as attachment.

Reviewer #2: The authors present FastTrack as a general tool to track any kind of objects in 2D and in a large panel of imaging conditions. My impression is that FastTrack is a very powerful tool for objects detection, but the tracking approach is weak in automatically solving occlusions, which is the very core of all tracking algorithms. The authors also present a database of different data on which the algorithm is tested. The algorithm is heavily based on manual checks through post-processing procedure, which is another weakness of the algorithm strictly related to the matching method followed by the authors.

My suggestion is to dampen the part of the manuscript in which the algorithm is defined as general and applicable to any type of data. My main concerns are indeed about the matching module of the algorithm and about the database.

Matching.

1- hard cost. the authors define h_ij as a hard cost, but in my opinion h_ij is instead a boolean variable associated to the pair i,j: h_ij=1 if r_ij < h_d otherwise it is equal to 0, where r_ij is the distance between i and j, and h_d is a threshold value. Therefore h_ij is a sort of participation variable that defines which pairs have a finite cost.

2- soft cost. It is not clear to me how s_d and s_a are chosen. The authors state that these 2 factors are the typical change in position and direction of motion respectively, but they do not give details on how these two factors are computed from the data, and my impression is that s_d and s_a are normalization factors that can be chosen from the users to increase the contribution of one term or the other in the soft cost definition.

3- occlusion. The matching between detected objects is performed using an Hungarian method, which can only produce one-to-one matches, which is clearly not a way to solve occlusions, but even worse it may produce swapping of identities. Video 1 of Supplementary Information clearly shows what happens when an occlusion occurs, as it is evident in the part of the movie with the 2 mice, but also in other parts of the video. While two objects are occluded the detected occluded detected object is not included in any trajectory and the output of the algorithm consists of 4 short trajectories that are matched before and after the occlusion. I guess that the identities are matched via a manual intervention of the user or via the part of the software related to the objects disappearance.

4- objects disappeareance. The authors should give more details on how the global cost matrix between not-matching objects is performed.

Post-processing.

The entire software is based on post-processing manual check of the algorithm output which is also used to train the matching procedure on data of the same type. My concern about this highly manual component of the algorithm is that this is applicable only on low density datasets of a small number of individuals.

Data.

1- public datasets. The authors state that the algorithm can be applied to a broad class of data. My question is then why they did not test the algorithm on already existing datasets, which is a common practise in the computer vision community.

2- tracking data. Many of the data included in the database are interesting from the object detection point of view, but they do not present particular difficulties from the tracking point of view. This is the case of the Drosophila examples in Video1 of the Supplementary Information, where Drosophila move in Y cells that do not communicate, hence tracking is essentially a segmentation and linking of detected objects within a specific region. This is also the case of the industrial examples of the same Video, where still objects are transported by small platforms.

Minor comments.

1- Fig.2c: how is the computational time normalized?

2- the authors should give the idea of the method chosen to extract the objects contours

3- in Section “Parameters optimization” the authors use the term swap to indicate an error in the matching, but swapping is not the only cause for a tracking error. A second kind of error is a missing match, which happens every time that an object is not included in any match. This last situation is what happens when h_d is too small and there are not possible matching candidates for the objects or for a part of them.

4- developped -> developed

Reviewer #3: The present paper introduces FastTrack, "an open-source software for tracking varying numbers of deformable objects". The authors

describe the software and it's workflow (mainly detection, matching, post processing). They introduce a replay feature, a system

to let users manually correct errors from automated tracking by swapping assigned identifiers. The authors provide a method to

estimate the number of manual corrections needed in a specific tracking task. Based on manually extracted ground truth, they also

introduce a method to optimize some of the parameters of the tracking algorithm.

* General Comments:

In general the paper is clear and well written, but it does not go deep into results, nor issues that can be encountered when

doing a tracking task. The state of the art is superficial.

It is very appreciated that the software is released under an open license, with documentation, and with datasets for testing it.

The software is operational and provides a rather well thought interface, although there are specific details that could be improved.

The probability of incursions P_inc may be one of the key contribution of the paper, unfortunately the authors did not connect

this value with empirical measurements of P_swap to assess the usefulness of their approach.

The authors show how P_swap can be used to optimize the tracking parameters. This approach is tedious and standard practice.

It is interesting to report some methodology, but there is no novelty in this, as far as I can see.

* Specific comments:

- Probability of incursions:

The approach is interesting, however explanations are scarce. In particular I could not fully understand Equation 3, with r left

undefined. The general idea that I could grasp is that based on estimated motion (from tracking algorithm) and Voronoi cells, the

authors estimate the probability of incursions which may lead to tracking issues (erroneous objects assignments). The authors

provide calculations of P_inc they made on several test videos, but they provide no reference to understand how P_inc may be

useful. A simple solution would be to manually extract P_swap value of all the videos and compare P_inc to P_swap. This would

already help understanding if this estimator is relevant.

- Computational efficiency of FastTrack

The authors claim that their software is fast, with no data to support the claim. Please do a benchmark, with a well characterized

hardware, compare to other softwares doing the exact same task.

- Details of tracking algorithm

While reading the paper, it was not clear to me how the software handles assignment of newly appeared objects, or how intermittently

appearing objects are handled.

- Ergonomics of the software

As I tested FastTrack, I had difficulties zooming in particular areas of the image, the authors might want to improve this aspect.

Also, the replay section could see some ergonomics improvements. Manual corrections are best carried out in non linear editing mode,

with simple controls to go back and forth between frames. I found only one slider to move between frames, and keyboard shortcuts

would alternate between translating the image and skipping frames. I also tried to do some manual corrections on a personal video

of moving fish, but my attempt was rather unsuccessful. As an example, I faced the case of 3 fish getting close together and the

overlay of ids and ellipses made it unclear where each fish was. Moreover, I don't see how to do manual corrections if some

objects are not detected during several frames and assignments jump to distant locations (for instance when 3 fish overlap in the

video during several frames). It may help to visualize past and future trajectories, and be able to select and drag a detected

object to its "real" location, and optionally to recalculate the trajectories of nearby objects in case of a 2-objects

swap. Finally, the software is extracting all the frames of analyzed videos in png format which might take a long time, or be

impossible in case of high resolution or long movies (example: 4K 24h @ 1fps could easily yield 260GB of images)

- Usefulness of the software

There are no tangible results about the tracking performance of the software. I would expect some measurements of speed, of

accuracy compared to ground truth, of ergonomics to manually post process results. This could be done using the data set

provided. Also, the authors may compare their results to other existing software to better highlight their contribution.

**Have all data underlying the figures and results presented in the manuscript been provided?**

Reviewer #1: Yes

Reviewer #2: Yes

Reviewer #3: Yes

PLOS authors have the option to publish the peer review history of their article (what does this mean?). If published, this will include your full peer review and any attached files.

Reviewer #1: **Yes: **Hangjian Ling

Reviewer #2: No

Reviewer #3: **Yes: **Alexandre Campo
---

## [Decision Letter · Decision Letter 1]

10 Jan 2021

Dear Dr Candelier,

We are pleased to inform you that your manuscript 'FastTrack: an open-source software for tracking varying numbers of deformable objects' has been provisionally accepted for publication in PLOS Computational Biology.

Best regards,

Manja Marz

Software Editor

PLOS Computational Biology

Manja Marz

Software Editor

PLOS Computational Biology

Reviewer's Responses to Questions

**Comments to the Authors:**

Reviewer #1: After reviewing the revised manuscript, I consider that all my questions have been fully addressed. I have no further comments, and recommend the manuscript to be published.

Reviewer #3: The revised paper is now clearer, the authors responses to reviewers comments are detailed and very welcome.

Also the improvements brought to the paper and the software for this second revision are great and required

lots of efforts, this is very appreciated.

The comparison with existing software helps understand the benefits brought by this software. Some results are

not really surprising, as ID tracker is known to be slow (at least at some point in time); I would have been curious

to see other softwares benchmarked but this choice makes sense since ID tracker is popular, and benchmarking

is a tedious task. The shift of focus to P_inc is welcome, removing the requirement of ground truth is a step forward.

Overall, the automation of parameterization, together with an interface to perform manual adjustments and corrections

gives a very useful tool to the communities of scientists that need effective video tracking.

**Have all data underlying the figures and results presented in the manuscript been provided?**

Reviewer #1: Yes

Reviewer #3: Yes

PLOS authors have the option to publish the peer review history of their article (what does this mean?). If published, this will include your full peer review and any attached files.

Reviewer #1: **Yes: **Hangjian Ling

Reviewer #3: **Yes: **Alexandre Campo

---

## [Editor Report · Acceptance letter]

4 Feb 2021

PCOMPBIOL-D-20-00512R1 

FastTrack: an open-source software for tracking varying numbers of deformable objects

Dear Dr Candelier,

I am pleased to inform you that your manuscript has been formally accepted for publication in PLOS Computational Biology. Your manuscript is now with our production department and you will be notified of the publication date in due course.

With kind regards,

Alice Ellingham
